# Relapse Patterns and Tailored Treatment Strategies for Malignant Pleural Mesothelioma Recurrence after Multimodality Therapy

**DOI:** 10.3390/jcm10051134

**Published:** 2021-03-08

**Authors:** Alice Bellini, Andrea Dell’Amore, Stefano Terzi, Giovanni Zambello, Andrea Zuin, Giulia Pasello, Fiorella Calabrese, Marco Schiavon, Federico Rea

**Affiliations:** 1Thoracic Surgery Division, Department of Cardiac, Thoracic, Vascular Sciences and Public Health, University of Padova, 35128 Padova, Italy; alicebellini26@gmail.com (A.B.); ste.terzi@gmail.com (S.T.); giovanni.zambello@aopd.veneto.it (G.Z.); andrea.zuin@unipd.it (A.Z.); marco.schiavon@unipd.it (M.S.); federico.rea@unipd.it (F.R.); 2Medical Oncology, Veneto Institute of Oncology IOV IRCCS, 35128 Padova, Italy; pasello@iov.veneto.it; 3Pathology Division, Department of Cardiac, Thoracic, Vascular Sciences and Public Health, University of Padova, 35128 Padova, Italy; fiorella.calabrese@unipd.it

**Keywords:** mesothelioma, thoracic surgery, multimodality therapy

## Abstract

To date, there have been no established therapies for recurrent malignant pleural mesothelioma (MPM) after multimodality treatment. Aims of this retrospective study are to analyze the recurrence pattern, its treatment and to identify the predictors of best oncological outcomes for relapsed MPM, comparing extrapleural pneumonectomy (EPP) vs. pleurectomy/decortication (PD). Study population: 94 patients with recurrence of MPM after multimodality treatment underwent macroscopic complete resection (52.1% with EPP and 47.9% with PD) between July 1994 and February 2020. Distant spread was the most frequent pattern of recurrence (71.3%), mostly in the EPP group, while the PD group showed a higher local-only failure rate. Post-recurrence treatment was administered in 86.2%, whereas best supportive care was administered in 13.8%. Median post-recurrence survival (PRS) was 12 months (EPP 14 vs. PD 8 months, *p* = 0.4338). At multivariate analysis, predictors of best PRS were epithelial histology (*p* = 0.026, HR 0.491, IC95% 0.263–0.916), local failure (*p* = 0.027, HR 0.707, IC95% 0.521–0.961), DFS ≥ 12 months (*p* = 0.006, HR 0.298, IC95% 0.137–0.812) and post-recurrence medical treatment (*p* = 0.046, HR 0.101, IC95% 0.897–0.936). The type of surgical intervention seems not to influence the PRS if patients are fit enough to face post-recurrence treatments. In patients with a prolonged disease-free interval, in the case of recurrence the most appropriate treatment seems to be the systemic medical therapy, even in the case of local-only relapse.

## 1. Introduction

Malignant pleural mesothelioma (MPM) is an aggressive asbestos-related tumor with poor prognosis. To date, multimodality treatment including chemotherapy and surgery, with or without radiotherapy, is the gold standard therapy for selected patients with epithelial and early stage MPM [1]. In this setting, the goal of surgery is to achieve the macroscopic complete resection (MCR) [2], obtained by either extrapleural pneumonectomy (EPP) or pleurectomy/decortication (PD). Failure, in local and/or distant sites, is one of the major concerns; in fact, there has been no established treatment for recurrence of MPM after the multimodality approach. Post-recurrence outcomes after EPP [3,4,5,6] and PD [7] have been reported. However, to the best of our knowledge, only one study [8] explored post-recurrence outcomes comparing EPP and PD in a multimodality setting.

This study aims to analyze the recurrence pattern, its treatment and to identify the predictors of best oncological outcomes for relapsed MPM after multimodality treatment, including both surgical procedures.

## 2. Material and Methods

Between July 1994 and February 2020, 250 patients were surgically treated with a multimodality protocol for MPM at Padova University Hospital. Patients with incomplete macroscopic resection (*n* = 36) and missing information about MCR (*n* = 2) were excluded from this retrospective study. All subjects gave their informed consent for inclusion before they participated in the study.

At the end of the follow up, 175 (82.5%) out of the 212 patients analyzed for relapse were found to have recurrent disease. Subsequently, we ruled out one patient for non-cancer-related death and 80 patients for missing data. In particular, we retrospectively included in the study only patients with unmistakable information about recurrence (both pattern of failure and treatments), and most of them were followed up at our institution. We included patients followed up by local oncologists only when the aforementioned information was completely available by phone interviews.

The final population was composed of 94 patients with recurrence of MPM after multimodality treatment (Figure 1).

Demographics (age at surgery and sex) and all relevant pre- and postoperative variables (histology, side, preoperative Charlson Comorbidity Index—CCI, preoperative Eastern Cooperative Oncology Group performance status—ECOG-PS, preoperative pulmonary function tests—PFTs, type of surgical intervention, pericardium and/or diaphragm resection with or without reconstruction, pT, pN, pathological stage, induction chemotherapy, adjuvant chemotherapy and radiotherapy, type of multimodality approach, pattern of failure, relapse treatment) were collected in order to identify possible prognostic factors. Clinical staging was based on total body computed tomography (CT) scan and positron emission tomography (PET) CT scan. The 8th edition of the lung cancer tumor, node and metastasis (TNM) staging system was used to define the extent of the disease [9].

### 2.1. Multimodality Treatment

Eligibility criteria for multimodality treatment included biopsy-proven MPM (of any histological subtype) at clinical stage T1-3 N0-2 M0 and anticipated complete resectability by EPP or PD, as estimated by an experienced thoracic surgeon in a multidisciplinary setting.

For induction and/or adjuvant chemotherapy, a platinum-based regimen with gemcitabine or pemetrexed was used for three to four cycles.

At our institution, we prefer to perform induction chemotherapy; in fact, according to our experience (a) it can be administered with high dosage in patients no longer debilitated by surgery; (b) it can lead to a down-staging of the disease, allowing to obtain a satisfactory macroscopic complete resection; (c) it allows for a better surgical selection based on the response to chemotherapy—a poor response may avoid an unnecessary surgical treatment in a more aggressive disease; (d) a high dose of adjuvant radiotherapy, particularly after extrapleural pneumonectomy, may be delivered, avoiding the cumulative toxicity.

Conversely, we performed the upfront surgery in very select cases: earlier clinical stage, with a very thin parietal pleura thickness and without visceral pleural and lymph nodes involvement.

Surgery was performed within 4–6 weeks of completing the final cycle of chemotherapy in patients who achieved at least a stable disease at CT scan and PET/CT scan.

In our hospital, EPP was the routine procedure proposed for MPM until 2012, becoming the main operative method when PD was introduced. Since then, conversion from PD to EPP was employed in the case of macroscopic pulmonary parenchyma invasion and interesting fissures. Resection and reconstruction of the pericardium and/or diaphragm were performed only in the case of macroscopic involvement.

Adjuvant radiotherapy was based on helical tomotherapy or intensity-modulated radiation therapy (dose range 40–50 Gy).

### 2.2. Post-Operative Follow-Up, Diagnosis and Treatment of Recurrence

Patients were followed up with clinical visits, imaging studies (CT scan and/or PET/CT scan) and phone interviews every 4–6 months. The diagnosis of recurrence was usually based on radiological features (CT and/or PET/CT scans) and was associated with histological or cytological analysis in a few unclear cases.

Local recurrence was defined as tumor relapse in the ipsilateral hemithorax, including chest wall, diaphragm, pericardium and ipsilateral lymph nodes (mediastinal, axillary, sub- and supraclavicular ones). Distant recurrence was defined as tumor recurrence in the contralateral hemithorax, abdomen or at other distant locations. Decisions about relapse treatment were decided by a multidisciplinary team, considering the recurrence pattern and the conditions of the patients.

In our analysis, we considered the cumulative pattern of relapse in order to compare MPM with iterative local recurrences, despite local and/or systemic therapy, with MPM with distant spread associated with local relapse or not. Consequently, we regarded global treatments for recurrences.

### 2.3. Statistical Analysis

The data were reported as absolute numbers, percentages or median values with interquartile range (IQR). The association between qualitative variables was verified by the Fisher test.

Overall survival (OS) was calculated from the date of surgery until that of death or last follow-up. Disease-free survival (DFS) was calculated as the time between surgery and diagnosis of recurrence. Post-recurrence survival (PRS) was calculated from the date of recurrence to the date of death or last follow-up. A survival analysis was performed by applying the Kaplan–Meier and Cox-regression methods. *p* < 0.05 was considered statistically significant.

All the statistical analyses were performed using SPSS 20.0 version for windows (SPSS, Inc., Chicago, IL, USA) and GraphPad Prism 8 Version 8.2.1 for macOS.

## 3. Results

### 3.1. Patient Characteristics

Patient characteristics are summarized in Table 1. EPP was performed in 49 (52.1%) cases, while PD was performed in 45 (47.9%). The groups were homogeneous for sex, histology (epithelial vs. non-epithelial), side, CCI, ECOG-PS, PFTs, diaphragm resection (no vs. yes), pT (complete remission/1/2 vs. 3/4), pN, *p*-stage (early vs. advanced), induction and adjuvant chemotherapy, adjuvant radiotherapy, multimodality treatments and post-recurrence treatment administration. Conversely, the EPP group included younger patients (*p* = 0.0026), lower scintigraphy perfusion in the pathological lung (*p* = 0.0240), greater need of pericardial resection (*p* < 0.0001), longer DFS (*p* = 0.0360) and local failure only (*p* = 0.0067). Moreover, when stratified for years, we observed a statistical difference in the type of surgical intervention employed; EPP was the routine procedure for MPM until 2012 when PD was introduced, which became the main operative method (*p* < 0.0001).

### 3.2. Pattern and Treatment of Relapse

Patterns of failure were local in 27 (28.7%) cases, distant in 27 (28.7%) and local and distant in 40 (42.6%). Particularly, recurrences were localized in the ipsilateral hemithorax in 27 (28.7%) patients, contralateral hemithorax in 18 (19.1%), abdomen in 11 (11.7%), ipsi- and/or contralateral hemithorax and abdomen in 26 (27.7%) and other distant sites with or without thorax and/or abdomen involvement in 12 (12.8%). In the EPP group (*n* = 49), relapse was local in 8 (16.3%) cases, distant in 23 (46.9%) and local and distant in 18 (36.7%). In the PD group (*n* = 45), relapse was local in 19 (42.2%) cases, distant in 4 (8.9%) and local and distant in 22 (48.9%).

Of the 94 patients with MPM recurrence, 81 (86.2%) patients received post-recurrence treatment, whereas 13 (13.8%) received best supportive care only (4 patients in the EPP group vs. 9 in the PD group), because of poor performance status (*n* = 7), rapid progressive disease (*n* = 2) or unknown cause (*n* = 4). Treatments included medical therapies (chemotherapy, radiotherapy, ongoing trails) (*n* = 68, 72.3%; 35 patients in EPP group vs. 33 in PD group) and redoing surgery (*n* = 13, 13.8%; 10 patients in EPP group vs. 3 in PD group). Particularly, a single treatment based on chemotherapy was administered in 48 (51.1%) cases, radiotherapy in 1 (1.1%), surgery in 3 (3.2%) and experimental therapies in 3 (3.2%), while 26 (27.7%) patients received different combinations of the aforementioned treatments (Table 2).

Chemotherapy and radiotherapy were globally administered in 74 (78.7%) and 14 (14.9%) patients, respectively. Specifically, second line chemotherapy based on rechallenging with platinum agent plus pemetrexed was administered in 27 cases, vinorelbine in 12, gemcitabine in 5, gemcitabine plus NGR-hTFNa vs. placebo (ongoing phase II trial) in 2, platinum agent plus raltitrexed in 1, while in 27 cases data about regimens were not available. Eleven (11.7%) patients were treated with experimental therapies.

Post-recurrence surgical treatment was performed in 13 patients: eight had resection of a single solid metastasis localized in the soft tissues of the ipsilateral chest wall (*n* = 3), ipsilateral pleura (*n* = 2), abdomen (*n* = 1), contralateral cheek (*n* = 1) and ipsilateral axillary lymphadenopathy (*n* = 1); two had open wedge resection for contralateral pulmonary relapse; one had ipsilateral mastectomy and one had peritonectomy and hyperthermic intraperitoneal chemotherapy for a diffuse abdominal failure. No complications occurred after the redo surgery.

### 3.3. Survival Outcomes and Prognostic Factors

Follow-up was completed on all the patients, with a median of 26.7 months (range 4–239 months), in particular 35.6 months (range 6–239 months) for the EPP group and 21.1 months (range 4–89 months) for the PD group.

Median OS for all patients was 33 months, while for the EPP and PD group it was 38 and 23 months, respectively (*p* = 0.0199) (Figure 2). The one-, two- and five-year OS rates were 93.3%, 75.6% and 31.1%, respectively, for the EPP group and 71.4%, 42.8% and 8.2%, respectively, for the PD group. According to the Cox-regression analysis, predictors of better OS were EPP (*p* = 0.011, HR 0.524, IC95% 0.318–0.863), epithelial histology (*p* = 0.001, HR 0.341, IC95% 0.182–0.639), tri-modality treatment (*p* = 0.012, HR 0.419, IC95% 0.212–0.826) and induction chemotherapy administration (*p* = 0.017, HR 0.151, IC95% 0.032–0.711).

Median DFS for all patients was 14 months (Figure 3), while for the EPP and PD group it was 20 and 11 months, respectively (*p* < 0.0001). The one-, two- and five-year DFS rates were 71.1%, 48.9% and 22.2%, respectively, for the EPP group and 42.9%, 12.2% and 2%, respectively, for the PD group. According to the Cox-regression analysis, predictors of better DFS were EPP (*p* = 0.001, HR 0.446, IC95% 0.281–0.708), epithelial histology (*p* = 0.02, HR 0.489, IC95% 0.268–0.893), tri-modality treatment (*p* = 0.001, HR 0.350, IC95% 0.189–0.649), pathological stage I (*p* = 0.015, HR 0.727, IC95% 0.563–0.939), induction chemotherapy (*p* = 0.012, HR 0.165, IC95% 0.040–0.672), adjuvant chemotherapy (*p* = 0.007, HR 0.302, IC95% 0.1260.726) and adjuvant radiotherapy (*p* = 0.004, HR 0.424, IC95% 0.238–0.755) administrations.

Median PRS for all patients was 12 months, while for the EPP and PD group it was 14 and 8 months (*p* = 0.4338), respectively (Figure 4). According to the Cox-regression analysis, predictors of better PRS were epithelial histology (*p* = 0.026, HR 0.491, IC95% 0.263–0.916), local failure (*p* = 0.027, HR 0.707, IC95% 0.521–0.961), DFS ≥ 12 months (*p* = 0.006, HR 0.298, IC95% 0.137–0.812) and post-recurrence medical treatment (*p* = 0.046, HR 0.101, IC95% 0.897–0.936).

The Cox-regression analysis results are summarized in Table 3.

## 4. Discussion

Recurrence of MPM after multimodality treatment is a common problem. Nevertheless, there has been no established therapy for relapse to date. Major studies about the treatment of recurrent MPM are reported in Table 4.

To our knowledge, this is the largest study to explore post-recurrence outcomes including EPP and PD in a multimodality setting. We assessed 94 patients who had recurrence after EPP (*n* = 49) and PD (*n* = 45) with the aim to analyze the recurrence pattern and its treatment and to identify the predictors of the best oncological outcomes for relapsed MPM after multimodality therapy.

In our study, MPM with distant spread (associated or not with local relapse) was the most frequent pattern of recurrence (71.3%), mostly in the EPP group (EPP group 83.6% vs. PD group 57.8%), while the PD group showed a higher local-only failure rate (EPP group 16.3% vs. PD group 42.2%). These data are in line with the literature, as shown in Table 4. We found a longer DFS for patients who underwent MCR with EPP compared to PD at 20 and 11 months, respectively, leading us to hypothesize that early local-only failure may represent a consequence of a less local radical resection, with a higher microscopic local persistent disease burden. In fact, both surgical procedures are cytoreductive, but PD is a lung-sparing surgery involving the removal of parietal and visceral pleura, theoretically less locally radical when compared with EPP [10,11,12,13]. Cautiously, we could affirm that local-only failure tends to occur earlier than distant failure, hence it may not represent a true relapse of MPM after multimodality treatment, while distant spread is always a real recurrence of the disease. Likely the aforementioned encouraging results in the EPP group are due to the higher percentage of early pathological stage according to 8th TNM edition (I/II 71.4% vs. III/IV 28.6%) compared to Kai et al. (I/II 44.8% vs. III/IV 55.2%) [8]; in fact, we found advanced pathological stage as an independent predictor of worse DFS.

On the other hand, we noted that the type of surgical resection did not affect the PRS. Both groups were fit enough to receive post-recurrence treatments, leading long-term outcomes after relapse.

Previous reports have highlighted poor prognosis in patients with recurrent MPM after multimodality treatment, with median PRS after EPP ranging from 3 to 6.5 months [3,4,5,6,8], whereas encouraging PRS were reported after PD by Nakamura et al. and Kai and collaborators (14.4 and 20 months, respectively) [7,8]. In the present study, overall median PRS was 12 months and 14 and 8 months in the EPP and PD group, respectively.

In recent years [14,15], PD has become the method of choice in our institution as, whenever MCR is technically feasible, we try to preserve the lung. The EPP group represents, in fact, an historical cohort of highly selected patients mostly operated on before 2012 (38 before 2012 vs. 11 after 2012). Particularly, of the 45 patients treated before 2012, 84.4% underwent EPP, while of the 49 operated after 2012 only 22.5% did. Indeed, important selection bias must be considered; we offer EPP only to fit, young patients, while, mostly since 2012, we tend to perform PD also in advanced stage, especially in the case of higher CCI and ECOG PS scores and/or PFTs precluding pneumonectomy (III/IV *p*-stage 14.3% before 2012 vs. 34.2% after 2012). Perhaps this is the reason why in the present study patients undergoing EPP seem to have better survival outcomes and are fit enough to face relapse with medical treatments, in contrast with Kai and colleagues who demonstrated a higher likelihood of receiving chemotherapy following recurrence for the PD group with improved PRS compared to the EPP group [8].

The Cox-regression analysis revealed that PRS was independently predicted by post-recurrence treatment, DFS, site of relapse and histology. Across the literature, post-recurrence treatment is the main predictor of better PRS [4,7,8], in particular we found tailored medical therapies as the best strategy to face relapse, even in the case of local failure, in contrast with satisfactory PRS after redoing surgery, which was reported by Kostron et al. As we mentioned, early local-only failure in our series may likely reflect a less local radical resection occurring from timely systemic therapies, rather than redo surgery that is rarely radical in most of the cases. Regardless, we found local recurrence as a predictor of the longest PRS, maybe due to a less deleterious effect on performance status and, consequently, on survival compared with distant spread. As already stated in the literature [6,7], a long DFS (≥12 months) result was significantly associated with good survival, probably reflecting a slower tumor growth speed associated with a less aggressive recurrent disease. Furthermore, epithelial histology, besides predicting OS and DFS, resulted as a favorable prognostic factor for PRS, as already reported by Kai et al. as a trend at multivariate analysis (*p* = 0.065).

In our analysis, although there was a difference in the pathological stage of DFS, there was no difference in the OS and PRS. This finding must be carefully interpreted as this is a selected population of relapsed patients only, resulting as a potential bias in the OS analysis. Maybe the pathological stage I tended to relapse was later influenced by other factors, for example a more aggressive histological behavior; as we reported, histology in our analysis affected OS more than DFS and PRS. Conversely, among the relapsed earlier stages of MPM the pathological stage I seemed to have better DFS, confirming its strong prognostic role [1]. Among the aforementioned studies reported in Table 4, only Kai and collaborators analyzed the prognostic role of the pathological stage, finding similar results: earlier stages did not affect either the OS or the PRS, but they did not investigate the DFS [8].

The current study has several limitations. Firstly, it is a single-center retrospective study. In addition, a possible selection bias related to preoperative performance status was present in the EPP group, thus all but 4 patients (8.2%) were fit enough to face post-recurrence therapies. Certainly, in a retrospective study the selection bias represents a limitation, which, however, is not the case in clinical practice where it is likely that a correct selection of patients can modify the results even after such a destructive intervention as EPP. Perhaps it is not yet time to abandon this type of intervention in young, fit and carefully evaluated patients in a multidisciplinary setting [19]. Indeed, an important limitation of this study is the different follow-up period for each cohort (35.6 months for the EPP group vs. 21.1 months for PD group); in fact, almost all the cases prior to 2012 were done with EPP. However, both groups completed in median at least two years of follow-up with better encouraging outcomes for the EPP group. For the EPP group the median one- and two-year OS rates were 93.3% and 75.6% for the EPP group and 71.4% and 42.8% for the PD group, respectively; while the median one- and two-year DFS rates were 71.1% and 48.9% for the EPP group and 42.9% and 12.2% for the PD group, respectively. Ultimately, a longer and similar follow-up time is necessary to better compare the two groups. Moreover, MPM is a biologically diverse disease still not totally understood, which may explain the different patterns of relapse. Further studies will be necessary to confirm the current findings and to establish final criteria for the indication of second-line therapies, currently decided on a patient-by-patient basis.

## 5. Conclusions

In our experience, the completion of multimodal treatment in patients with epithelial histology is associated with better oncological outcomes, in particular earlier stages undergoing EPP tend to relapse later. On the contrary, the type of surgical intervention (EPP vs. PD) seems not to influence the PRS if patients are fit enough to face post-recurrence treatments. In patients with a prolonged disease-free interval, in the case of recurrence, the most appropriate treatment seems to be the systemic medical therapy, even in case of local-only relapse.

## Figures and Tables

**Figure 1 jcm-10-01134-f001:**
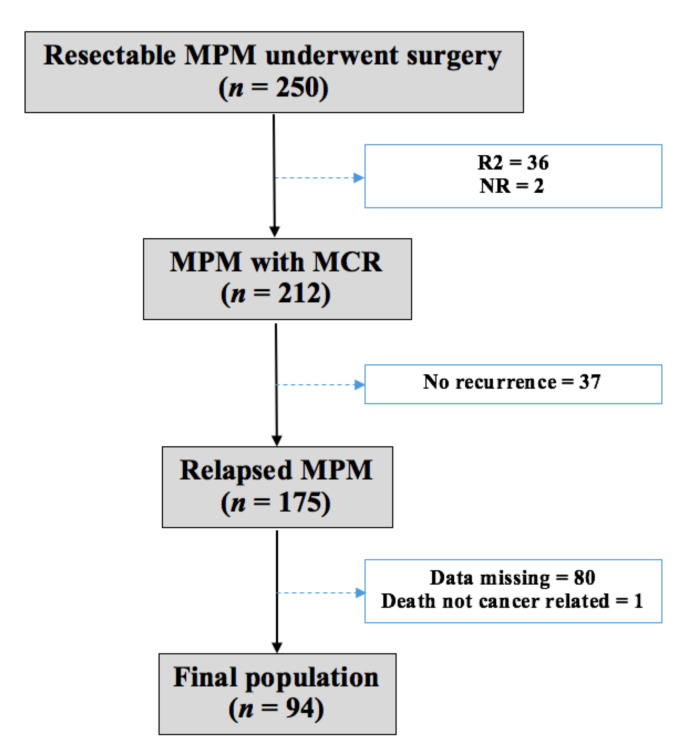
Population. MPM—malignant pleural mesothelioma; R2—incomplete macroscopic resection; MCR—macroscopic complete resection.

**Figure 2 jcm-10-01134-f002:**
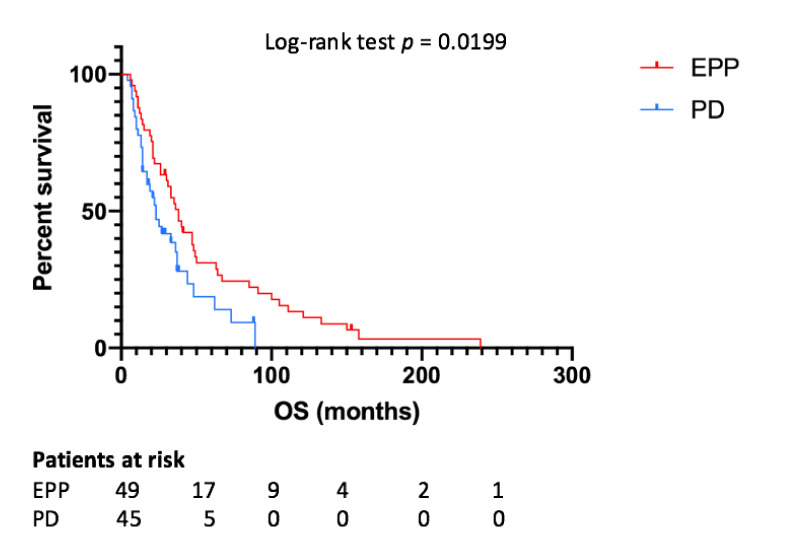
Overall survival depending on type of surgery. EPP—extrapleural pneumonectomy; PD—pleurectomy/decortication; OS—overall survival.

**Figure 3 jcm-10-01134-f003:**
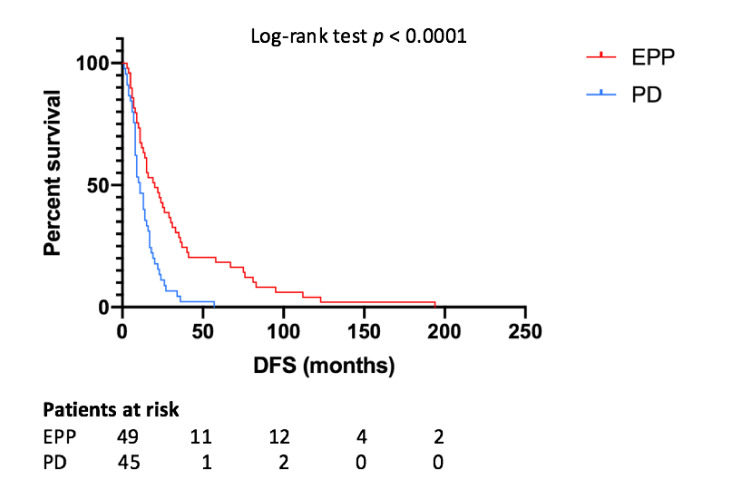
Disease-free survival depending on type of surgery. EPP—extrapleural pneumonectomy; PD—pleurectomy/decortication; DFS—disease-free survival.

**Figure 4 jcm-10-01134-f004:**
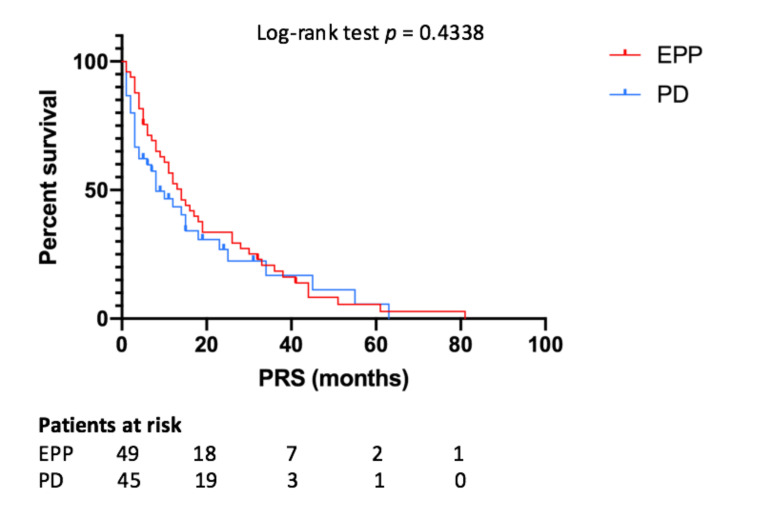
Post-recurrence survival depending on type of surgery. EPP—extrapleural pneumonectomy; PD—pleurectomy/decortication; PRS—post-recurrence survival.

**Table 1 jcm-10-01134-t001:** Characteristic of patients (*n* = 94).

Characteristics	All Patients *n* = 94	EPP Group *n* = 49	PD Group *n* = 45	*p*-Value
**Sex (male: female), *n* (%)**	69:25 (73.4:26.6)	38:11 (77.5:22.5)	31:14 (68.9:31.1)	0.3612
**Age at surgery (years), median (IQR)**	64.7 (58–70)	63 (58–68)	69 (62.5–72)	0.0026
**Histology, *n* (%)**				0.5571
Epithelial	81 (86.2)	41 (83.7)	40 (88.9)	
Sarcomatous	1 (1.1)	1 (2)	0 (0)	
Biphasic	10 (10.6)	6 (12.3)	4 (8.9)	
Desmoplastic	2 (2.1)	1 (2)	1 (2.2)	
**Side, *n* (%)**				0.3989
Right	58 (61.7)	28 (57.1)	30 (66.7)	
Left	36 (38.3)	21 (42.9)	15 (33.3)	
**CCI, *n* (IQR)**	4 (3–7)	4 (3–7)	4 (4–7)	0.0745
**ECOG-PS**				>0.9999
0	57	29	28	
1	29	14	15	
**FEV1%, *n* (IQR)**	77 (69–89)	80 (69–88.5)	76.5 (68.75–91.25)	0.7624
**FVC%, *n* (IQR)**	77 (64–88)	76 (62–81.5)	79 (66.25–89.75)	0.2386
**VC%, *n* (IQR)**	79 (67–87)	79 (62.5–87)	79.5 (69.75–92.25)	0.4865
**TLC%, *n* (IQR)**	79 (72–89)	79 (72–88.5)	79.5 (70.75–90.5)	0.6916
**DLCO%, *n* (IQR)**	68 (57.5–77)	64 (52–75)	69 (61–80)	0.1386
**VO2 max (ml/kg/min), *n* (IQR)**	17.3 (15.88–21.1)	17.15 (15.83–21.38)	17.3 (15.88–21.1)	0.933
**Pathological lung scintigraphy perfusion%, *n* (IQR)**	33.68 (26.82–39.38)	29 (25.69–37)	37.38 (29–43.64)	0.024
**Pathological lung scintigraphy ventilation%, *n* (IQR)**	30.4 (20.5–38)	28.94 (16.92–34.07)	34.7 (22.95–44.39)	0.0675
**Surgical intervention, *n* (%)**				<0.0001
Before 2012	45 (47.9)	38 (77.6)	7 (15.6)	
After 2012	49 (52.1)	11 (22.5)	38 (84.4)	
**Pericardium resection, *n* (%)**				<0.0001
No	20 (21.3)	2 (4.1)	18 (40)	
Yes	3 (3.2)	1 (2)	2 (4.4)	
Yes, reconstruction with patch	71 (75.5)	46 (93.9)	25 (55.6)	
**Diaphragm resection, *n* (%)**				0.4733
No	8 (8.5)	3 (6.1)	5 (11.1)	
Yes, direct suture	5 (5.3)	0 (0)	5 (11.1)	
Yes, reconstruction with patch	81 (86.2)	46 (93.9)	35 (77.8)	
**pT (TNM VIII edition), *n* (%)**				0.1921
Complete remission	2 (2.1)	1 (2)	1 (2.2)	
1	5 (5.3)	1 (2)	4 (8.9)	
2	24 (25.5)	11 (22.5)	13 (28.9)	
3	45 (47.9)	28 (57.1)	17 (37.8)	
4	18 (19.1)	8 (16.3)	10 (22.2)	
**pN (TNM VIII edition), *n* (%)**				>0.9999
0	71 (75.5)	37 (75.5)	34 (75.5)	
1	23 (24.5)	12 (24.5)	11 (24.5)	
**Pathological stage (TNM VIII edition), *n* (%)**				0.8244
Complete remission	2 (2.1)	1 (2)	1 (2.2)	
I	57 (60.6)	32 (65.3)	25 (55.6)	
II	7 (7.4)	2 (4.1)	5 (11.1)	
III	13 (13.8)	8 (16.3)	5 (11.1)	
IV	15 (16)	6 (12.3)	9 (20)	
**Induction chemotherapy, *n* (%)**				0.2433
No	3 (3.2)	3	0 (0)	
Yes	91 (96.8)	46	45 (100)	
**Adjuvant chemotherapy, *n* (%)**				0.2537
No	87 (92.5)	47 (95.9)	40 (88.9)	
Yes	7 (7.5)	2 (4.1)	5 (11.1)	
**Adjuvant radiotherapy, *n* (%)**				0.1592
No	15 (16)	5 (10.2)	10 (22.2)	
Yes	79 (84)	44 (89.8)	35 (77.8)	
**Multimodality treatment, *n* (%)**				0.6012
Bimodal	18 (19.1)	8 (16.3)	10 (22.2)	
Trimodal	76 (80.9)	41 (83.7)	35 (77.8)	
**DFS, *n* (%)**				0.036
<12 months	39 (41.5)	15 (30.6)	24 (53.3)	
≥12 months	55 (58.5)	34 (69.4)	21 (46.7)	
**Local failure only, *n* (%)**				0.0067
No	67 (71.3)	41 (83.7)	26 (57.8)	
Yes	27 (28.7)	8 (16.3)	19 (42.2)	
**Post-recurrence treatment, *n* (%)**				0.1361
No	13 (13.8)	4 (8.2)	9 (20)	
Yes	81 (86.2)	45 (91.8)	36 (80)	

EPP—extrapleural pneumonectomy; PD—pleurectomy/decortication; CCI—Charlson comorbidity index; ECOG-PS—Eastern Cooperative Oncology Group performance status; FEV1%—forced expiratory volume in 1 s; TLC%—total lung capacity; DFS—disease free survival.

**Table 2 jcm-10-01134-t002:** Recurrence pattern and treatment.

	*n* (%)
**Recurrence pattern**	
Local	27 (28.7)
Distant	27 (28.7)
Local + distant	40 (42.6)
**Recurrence site**	
Ipsilateral hemithorax	27 (28.7)
Contralateral hemithorax	18 (19.1)
Abdomen	11 (11.7)
Thorax + abdomen	26 (27.7)
Others	12 (12.8)
**Recurrence treatment**	
None	13 (13.8)
CT	48 (51.1)
RT	1 (1.1)
Surgery	3 (3.2)
CT + Surgery	6 (6.4)
CT + RT	9 (9.6)
CT + RT + Surgery	3 (3.2)
CT + Other	7 (7.4)
CT + RT + Other	1 (1.1)
Other	3 (3.2)

CT—chemotherapy; RT—radiotherapy.

**Table 3 jcm-10-01134-t003:** Predictors of better survival outcomes at multivariate Cox-regression analysis.

	*p*-Value	HR	IC95%
**OS**			
EPP	0.011	0.524	0.318–0.863
Epithelial histology	0.001	0.341	0.182–0.639
Trimodality treatment	0.012	0.419	0.212–0.826
Induction chemotherapy	0.017	0.151	0.032–0.711
**DFS**			
EPP	0.001	0.446	0.281–0.708
Epithelial histology	0.02	0.489	0.268–0.893
Trimodality treatment	0.001	0.35	0.189–0.649
Pathological stage I	0.015	0.727	0.563–0.939
Induction chemotherapy	0.012	0.165	0.040–0.672
Adjuvant chemotherapy	0.007	0.302	0.126–0.726
Adjuvant radiotherapy	0.004	0.424	0.238–0.755
**PRS**			
Epithelial histology	0.026	0.491	0.263–0.916
Local failure	0.027	0.707	0.521–0.961
DFS ≥ 12 months	0.006	0.298	0.137–0.812
Post-recurrence medical treatment	0.046	0.101	0.897–0.936

OS—overall survival; EPP—extrapleural pneumonectomy; DFS—disease-free survival; PRS—post recurrence.

**Table 4 jcm-10-01134-t004:** Major studies about the treatment of recurrent MPM.

Author	Surgery, *n*	Multimodality, *n*	Relapse, *n* (%)	Pattern of Recurrence, %	Median DFS (m)	Relapse Treatment, *n* (%)	Median PRS (m)	Median OS (m)
Kostron, 2015 [4]	EPP, 136	Bimodal, 47 Trimodal, 59	106 (77.9)	L 24.3 D 19.9 L + D 33.8	9	None, 28 (26.4) Surgery, 16 (15.1) Medical treatment, 73 (68.9)	7	22 ^b^
Takuwa, 2017 [5]	EPP, 59	Bimodal, 27 Trimodal, 12	39 (66.1)	NR	11.6	None, 12 (30.7) Medical treatment, 27 (69.2)	6.5	22
Kai, 2018 [8]	EPP, 29 PD, 15	Bimodal, 26 Trimodal, 18	32 (72.7)	L 18.2 D 27.3 L + D 27.3	Overall, 14 ^c^ EPP, 13 ^c^ PD, 21 ^c^	Medical treatment, 17 (53.1)	Overall, 5 EPP, 3 PD, 20	Overall, 22 ^c^ EPP, 17 ^c^ PD, 34 ^c^
Soldera, 2019 [6]	EPP, 93	Bimodal 43 Trimodal 10	53 (57.0)	L 5.4 D 38.7 L + D 12.9	NR	None, 27 (50.9) Medical treatment, 15 (28.3) NR, 11 (20.8)	4.8	NR
Nakamura, 2020 [7]	PD, 90	Bimodal, 90	57 (63.3)	L 43 D 6.7 L + D 13.3	19	Surgery, 3 (5.3) Medical treatment, 40 (70.2) Best supportive care, 14 (24.5)	14.4	57
Politi, 2010 [16]	EPP, 8	NR	8 (100)	L 50 D 50	NR	Surgery, 8 (100)	14.5	NR
Okamoto, 2013 [17]	EPP, 10	NR	8 (80)	L 40 D 40	15.4	Surgery, 2 (25) Medical treatment, 6 (75)	17.8	49.6 ^a^
Burt, 2012 [18]	EPP, 32 PD, 15	NR	47 (100)	L 100	16.1	Surgery, 47 (100)	Epithelial, 20.4 Biphasic, 7.0	44.9
Present series	EPP, 49 PD, 45	Bimodal, 18 Trimodal, 76	94 (100)	L, 28.7 D, 28.7 L + D, 42.6	Overall, 14 EPP, 20 PD, 11	None, 13 (13.8) Surgery, 13 (13.8) Medical treatment, 68 (72.3)	Overall, 12 EPP, 14 PD, 8	Overall, 33 EPP, 38 PD, 23

^a^ From initial treatment; ^b^ from the first cycle of neoadjuvant CT; ^c^ from the date of diagnosis. MPM—malignant pleural mesothelioma; EPP—extrapleural pneumonectomy; PD—pleurectomy/decortication; NR—not reported; L—local; D—distant; L + D—local + distant; DFS—disease-free survival; PRS—post-recurrence survival; OS—overall survival.

## Data Availability

The data presented in this study are available on request from the corresponding author.

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
