# Peer review of "Relapse Patterns and Tailored Treatment Strategies for Malignant Pleural Mesothelioma Recurrence after Multimodality Therapy"

_jcm, 2021, doi:10.3390/jcm10051134_

Round 1

Reviewer 1 Report

Dear authors,

congratulations for this study and its really interesting results. You have investigated relapse patterns and treatment strategies for MPM recurrence in patients, who underwent multimodality therapy before. This is an interesting and in my opinion really clinically relevant question.

I find you study design acceptable. However, the absence of n=80 patients seems a bit strange, so please comment on this limitation in a bit more detail.

Nearly all of your patients received induction chemotherapy. Did you see more postoperative complications? Why did you perform induction chemotherapy? Had these patients extensive tumor disease or positive lymph nodes? We changed our protocol from induction to adjuvant chemotherapy, since we lost some patients for radical surgery (patients were not suitable for surgery due to reduced overall condition) after induction chemotherapy.

How did you define recurrence, only based on imaging or did you always take a biopsy for histological confirmation?

OS and DFS were significantly better after EPP compared to P/D. You have discussed this fact in the discussion. Then why do you continue to perform the P/D and not the EPP again? You could do additional local procedures like the HITOC after P/D. Please comment briefly on that.

Taken together, this study delivers interesting results, is well written and of interst for the thoracic community.

Author Response

Thank you for your careful review and advises to improve our work, herein attached the point by point replies and modifications based on your comments 

Reviewer 2 Report

I think the most important thing to consider in this analysis is the different follow-up period for each cohort.

All cases prior to 2012 were done with EPP, and long-term survival of EPP may have apparently extended MST in the EPP group. It needs to be considered as an important limitation.

In MMP, the pathological stage seems to be the strongest prognostic factor, as shown by the large-scale study. However, in this analysis, although there is a difference in the pathological stage of DFS, there is no difference in the OS. I wonder why.

I feel that there are many references to literature that support EPP.

It would be better to have comments on the following literature that supports PD. Ichiki Y, Goto H, Fukuyama T, Nakanishi K. Should lung-sparing surgery be the standard procedure for malignant pleural mesothelioma?

J Clin Med. 8,9 (7); 2153.doi: 10.3390 / jcm 9072153, 2020.

Author Response

Thank you so much for all the suggestion, advises and comments to improve our work, we appreciate so much, herein attached the authors replies

regards  

Round 2

Reviewer 2 Report

This manuscript has been very well modified. I think that this manuscript seems to be of quality worthy of publish.